# Assessing the Impact of Non-Exhaust Emissions on the Asthmatic Airway (IONA) Protocol for a Randomised Three-Exposure Crossover Study

**DOI:** 10.3390/ijerph21070895

**Published:** 2024-07-09

**Authors:** James Scales, Hajar Hajmohammadi, Max Priestman, Luke C. McIlvenna, Ingrid E. de Boer, Haneen Hassan, Anja H. Tremper, Gang Chen, Helen E. Wood, David C. Green, Klea Katsouyanni, Ian S. Mudway, Christopher Griffiths

**Affiliations:** 1Asthma and Lung UK Centre for Applied Research, Edinburgh EH10 5HF, UK; 2Wolfson Institute for Population Health, Barts and The London School of Medicine and Dentistry, Queen Mary University of London, London E1 2AB, UK; 3MRC Centre for Environment and Health, Imperial College London, London W12 0BZ, UK; 4NIHR Health Protection Research Unit in Environmental Exposures, Imperial College London, London W12 0BZ, UK

**Keywords:** exercise, traffic, air pollution, wheeze, public health

## Abstract

Background: People living with asthma are disproportionately affected by air pollution, with increased symptoms, medication usage, hospital admissions, and the risk of death. To date, there has been a focus on exhaust emissions, but traffic-related air pollution (TRAP) can also arise from the mechanical abrasion of tyres, brakes, and road surfaces. We therefore created a study with the aim of investigating the acute impacts of non-exhaust emissions (NEEs) on the lung function and airway immune status of asthmatic adults. Methods: A randomised three-condition crossover panel design will expose adults with asthma using a 2.5 h intermittent cycling protocol in a random order at three locations in London, selected to provide the greatest contrast in the NEE components within TRAP. Lung function will be monitored using oscillometry, fractional exhaled nitric oxide, and spirometry (the primary outcome is the forced expiratory volume in one second). Biomarkers of inflammation and airborne metal exposure will be measured in the upper airway using nasal lavage. Symptom responses will be monitored using questionnaires. Sources of exhaust and non-exhaust concentrations will be established using source apportionment via the positive matrix factorisation of high-time resolution chemical measures conducted at the exposure sites. Discussion: Collectively, this study will provide us with valuable information on the health effects of NEE components within ambient PM_2.5_ and PM_10_, whilst establishing a biological mechanism to help contextualise current epidemiological observations.

## 1. Introduction

The global burden of disease survey estimated that air pollution contributed to one in eight deaths in 2019 [1]. These effects are largely attributed to long-term exposures to fine particulate matter (PM), PM_2.5_ (particulate matter with an aerodynamic diameter of, on average, less than 2.5 μm in diameter) and nitrogen dioxide (NO_2_), but short-term exposures are also associated with significant health impacts in vulnerable groups, such as people with asthma, including the worsening of symptoms [2], increased hospitalisation [3,4], and death [5].

Traffic as a source of PM has received considerable attention [6], with evidence that distance to roads, traffic density/composition [7], and exhaust emission tracers, such as elemental/black carbon, particle number concentration, and NO_2_ [8], are all associated with both acute and long-term negative health effects. There is therefore a need to better understand the contribution of different components of TRAP to these adverse effects, both to provide better advise to vulnerable groups and to help shape government policy to reduce all traffic emissions. To date, the contribution of traffic-derived air pollution research has focused on the contribution of exhaust emissions, whilst particulates from NEEs have been understudied [9]. Particles arising from tyre and brake wear, as well as the resuspension of road dust, represent a greater proportion of roadside PM by mass than direct exhaust emissions [10]. As exhaust emissions decrease as nations strive to meet their NetZero commitments, greater attention on non-exhaust sources is urgently required to evaluate their relative hazards compared with other pollutant sources [11]. NEEs remain unregulated, and their health impacts underexplored, particularly in vulnerable groups. Air pollution has been shown to significantly impact people living with asthma, increasing the prevalence of childhood asthma [12], exacerbations in adults [13], and hospital admissions [14]. As such, research into the health effects of NEEs on people with asthma is urgently needed.

Disentangling the relative contributions of different chemical mixtures within ambient PM has proven challenging in both short- and long-term epidemiological studies, due to the high degree of correlation between components with a common source. Currently, there is no firm consensus as to which components of PM_2.5_ or PM_10_ (particulate matter with an aerodynamic diameter of, on average, less than 10 μm in diameter) present the greatest hazard to the population [15]. Previous work [16,17,18,19,20,21] has demonstrated the feasibility of tackling this challenge using a quasi-experimental human real-world exposure crossover design in healthy subjects. Whilst this work showed the key role played by primary gaseous and particulate emissions, the study focused on responses in healthy adults and did not examine the biological pathways that lead to symptom exacerbation in vulnerable populations such as in individuals with asthma. These studies examined a range of metals such as iron (Fe), copper (Cu), nickel (Ni), and vanadium (V), reflecting their capacity to cause oxidative stress, but did not explicitly relate these to NEEs, nor employ them for detailed source apportionment.

To date, whether real-world NEEs are causally related to the worsening of asthma has not been widely studied. Therefore, the primary aim of this study is to assess and compare the acute changes in lung function and airway inflammation in asthmatics when exposed to three microenvironments with contrasting non-exhaust traffic-related air quality profiles. The contrasting traffic profiles increase the contrast in PM_10_ and PM_2.5_ from brake wear, tyre wear, and road abrasion, although tyre and road wear are often combined into a single source of tyre and road wear particles (TRWP).

## 2. Methods and Analysis

### 2.1. Study Design

The IONA study will employ a randomised three-condition crossover panel design. We will recruit 48 adults (aged over 18 years) residing in London, with mild-to-moderate allergic asthma that started in childhood. Inclusion and exclusion criteria were chosen to ensure participant safety and reduce confounding effects and is presented in Table 1.

### 2.2. Aims/Outcomes

#### 2.2.1. Primary

To compare the acute health effects (changes in lung function and airway inflammation) in mild/moderate asthmatics of exposure to PM_2.5_ and PM_10_ at three selected microenvironments, with contrasting contributions from brake wear and tyre and road wear (TRWP).

#### 2.2.2. Secondary

The provision of an air pollutant database (PM_2.5_ and PM_10_ mass and chemical composition, PNC, and NO_2_) and a time series of source-apportioned PM_2.5_ and PM_10,_ covering all exposure days at the three selected sites.To examine the relationship between variations in daily non-exhaust source fractions derived from brake and TRWP, with the deposited dose, based on metal/metalloid concentrations determined in the nasal airways, relating these measures of biological doses to the physiologic and immunologic responses observed.To establish a biobank of plasma, nasal lavage, and urine samples for future work examining molecular signatures of exposure and responses to NEEs.

#### 2.2.3. Outcome Measures

Primary: Lung function as measured by the forced expiratory volume in one second (FEV_1_); this measure has been selected as the primary outcome based upon previous work showing the detrimental effects of the time spent in an urban atmosphere on lung function [22].Secondary: Spirometry (FVC, FVC/FEV_1_ ratio), Fractional Expired Nitric Oxide (FeNO), oscillometry (R5, R20, T5, T20, AX), asthma symptoms, nasal mucosal immune responses.Biobanks: urine and blood plasma samples will be collected for future analysis.

### 2.3. Study Interventions

Due to the complexity of multipollutant aerosol mixtures within exhaust and NEEs, we will utilise an efficient randomised cross-over semi-experimental design investigating short-term respiratory health impacts in non-smoking adults with mild–moderate asthma during and after sequential standardised exercise exposures to three contrasting air quality environments, comprising the following:(1)A busy road junction characterised by stop–go traffic to enhance emissions from brake wear;(2)High speed continuous traffic to enhance TRWP emissions;(3)An urban background location away from nearby traffic sources.

Sites have been selected to provide the greatest contrast in emission sources and to be a cost-effective method of establishing the relationship between PM source contributions and acute health effects within the real-world context.

The overall protocol, and the participant flow through the three separate exposures, are summarised in Figure 1, with the assessment and sampling time points presented in Table 2. Following consent and initial assessment, participants will visit the three field testing sites in a random order. During the exposure protocol, participants will ride on a static bike for 15 minutes at a time, with 15 min rest periods, for a total of six cycling periods at a standardised intermittent intensity. Participants will perform identical respiratory health assessments preceding, during, and following the exercise task. In parallel to exposures at the sites, we will perform real-time measurements of PM_10_ and PM_2.5_, PNC and gaseous pollutants ozone (O_3_), NO_2_, and black carbon. The protocol for the three exposure visits will be identical, and each visit will be separated by a minimum of a two-week washout period, with testing taking place between May and October 2023.

To control for variations in the air quality, exposure will occur between 11:00 a.m. and 13:30 p.m. on weekdays. Between the pre- and post-health assessments defined in Figure 2, participants will be asked to cycle on a static exercise bike at a watt equivalent to 60% of the estimated heart rate. This exercise intensity was chosen to maximise the inhaled doses of PM and gaseous pollutants over the exposure period while minimising the likelihood of ventilatory drift and minimising the discomfort for the participants. Physiological drift will be monitored using the Ratings of Perceived Exertion questionnaire and the real-time monitoring of the participant’s heart rate.

#### 2.3.1. Exposure Site Selection

Three sites will be used for the exposure visits. The two air quality supersites in London, located in a suburban park in South London (Honor Oak Park) and close to a major trunk road in Central London (Marylebone Road), will be used as a basis for this study. These sites are equipped with the ability to collect time-resolved chemical composition measurements to quantify the non-exhaust component of PM. An additional measurement location will be established below a high-speed flyover (A40 flyover) using a mobile measurement facility equipped with the same highly time-resolved chemical composition measurement capability as the supersites. This location provides the contrast in non-exhaust contributions, being next to the high speed A40 flyover (Figure 2).

The Marylebone Road station is located 2 m from kerbside; this major trunk road carries 51,000 vehicles per day—4% heavy goods vehicles (HGV), 3% buses, and 16% light goods vehicles (LGV)—and the average speed is 34 kph (21 mph) [23]. The stop–go traffic on this congested road increases brake wear relative to tyre wear and resuspension. This has been confirmed by the London Atmospheric Emissions Inventory, which estimates 51% of NEEs as brake wear, compared to 39% as resuspension and 10% as tyre wear.

The proposed location on the A40 flyover is 5 m from the kerbside of the busy A40 flyover, carrying 93,000 vehicles per day—6% HGVs and 17% LGVs—and the average speed was 33–59 kph (21–37 mph) [23]. This road is a major route into and out of Central London; the high number of HGV and associated vehicle weight will lead to increased resuspension and tyre wear [9,24]. Due to the higher speeds and more free-flowing traffic, the London Atmospheric Emissions Inventory estimates this location to have only 85% of the brake wear emissions of the Marylebone Road measurement location, thereby increasing the contrast between the two traffic locations.

#### 2.3.2. Sample Size

The study uses an efficient Williams design [25] to achieve a significance level of 0.05 and a Δ (difference between the reduction in FEV_1_ between two sites) of 4.69 with a standard deviation of 6.4 [22]. At 91.5% power, 30 participants are required. To account for a dropout rate of 30% and uncertainty regarding the exact exposures on study days, we intend to recruit 48 participants in total. Since we will randomise the participants to a sequence for their exposures to the three sites, we will have six sequences (ABC/BCA/CBA/ACB/BAC/CAB). We will define twelve groups of up to six individuals each and then assign a group to each participant using randomly numbered sheets sealed in opaque envelopes.

#### 2.3.3. Recruitment

##### Primary Care

A Participant Identification Centre site approach will be used between May and September 2023 to recruit participants. Initially, participants will be recruited from general practice offices across Central and East London. Eligible participants identified by searches will be approached with an SMS invitation to join the study via secure electronic mail.

##### Non-Primary Care

Participants will also be recruited through other avenues. We will target people with asthma who study or work at Queen Mary, the University of London, and Imperial College London. Posters, lecture visits, and mailing lists will be used. Finally, we will recruit participants from the public through social media and posters.

#### 2.3.4. Patient and Public Involvement

This study was developed with input from the Asthma and Lung UK Centre for Applied Research (AUKCAR) Public and Patient Involvement (PPI) group. The group contributed to the study via providing feedback on the study design and methods. As the study progresses, the PPI group will be invited to project management group meetings as advisors and partners to discuss the analysis, results, and dissemination.

### 2.4. Preliminary Assessment:

Prior to the assessment at the exposure sites, participants will perform an exercise test to calculate the exercise intensity (described below), perform pre- and post-bronchodilator spirometry, and complete an Asthma Control Test [26] to confirm their asthma status. The spirometry assessment is defined in the data collection and analysis section below.

### 2.5. Estimated VO_2max_ Fitness Assessment

To support the standardisation of exercise intensity at exposure sites, participants will perform an estimated VO_2max_ test during the initial assessment. Participants will be asked to perform an incremental exercise test following YMCA cycle ergometry protocols defined by The American College for Sports Medicine [27]. Results will be used to calculate the estimated VO_2max,_ which will in turn be used to calculate a target watt equivalent for the 60% estimated VO_2max_. The Vo_2_ and heart rate scores will be inspected to ensure a linear relationship during the exercise protocol to ensure the accuracy of the estimate of the exercise intensity.

### 2.6. Exposure

To control for variations in air pollution, exposures will be in groups of between 2 and 6 participants at 11 a.m. until 13:30 p.m. on weekdays. Participants will exercise on cycle ergometers (Wattbike Gen II, Wattbike Ltd., Nottingham, UK) at a corresponding power output of 60% of their estimated VO_2max_ for six 15 min exercise bouts, interspersed with 15 min rest periods. This intensity is chosen to minimise the cardiovascular drift and maximise the inhaled dose. The 15 min rest periods are chosen to minimise cardiovascular drift, to allow comparisons to previous laboratory-based work, and to facilitate multiple participants at the testing site. Exposure assessments will be conducted between May and October 2023.

### 2.7. Data Collection before and after Exposure

Before and after each exposure, a battery of physiological measurements and samples will be taken. The time points of each measurement or sample are presented in an overview in Figure 1.

### 2.8. Respiratory Function Assessment:

#### 2.8.1. Spirometry

Our primary outcome, FEV_1_, will be assessed using a portable desktop computerised spirometer (Vitalograph, Buckingham, UK)), according to European Respiratory Society [28] and Association for Respiratory Technology and Physiology guidelines [29]. All testing was performed by trained operators under the supervision of a senior respiratory physiologist.

#### 2.8.2. Fractional Exhaled Nitric Oxide

The fractional exhaled nitric oxide (FeNO) can be an indicator of airway inflammation. FeNO will be measured using NIOX VERO (Niox, Oxford, UK) in accordance with the manufacturer’s instructions and the American Thoracic Society guidelines [30].

#### 2.8.3. Oscillometry

Airway resistance (R_5_, R_20,_ and R_5_–R_20_) and reactance (X_5_ and AX) will be measured using an oscillometer (Tremoflo^®^ C-100, Thorasys, Montreal, QC, Canada) in accordance with manufacturer’s instructions and the European Respiratory Society guidelines [31].

### 2.9. Sample Collection

#### 2.9.1. Venepuncture

Blood samples will be obtained via venepuncture immediately before exposure and the day after exposure. A total of 10ml will be collected into EDTA coated Vacutainers at each sample time point. Samples will be centrifuged at 3000 RPM at 4 °C for 15 min (Thermo Scientific Heraeus Multifuge 3 Plus Centrifuge, Walthan, MA, USA). Plasma will then be aliquoted into clean polypropylene tubes and frozen at −80 °C for later analysis and biobanking.

#### 2.9.2. Nasal Lavage

Nasal lavage samples will be collected using the nebuliser spray method [32] at three time points as follows: pre-, immediately post-, and 24 h post-exposure. Samples will be centrifuged to remove mucus and cellular components at 800× *g* at 4 °C for 10 min (Thermo Scientific Heraeus Multifuge 3 Plus Centrifuge), as well as the cell-free supernatant aliquoted for longer-term storage at −80 °C until required for analysis, or for longer-term storage as part of the sample biobank.

These samples will be used to test the hypothesis that exposure to NEE components causes changes in the immune status of the airway mucosa, skewing it toward a more Th2, Th17 profile, consistent with an increased susceptibility of the airway to a range of asthma triggers. At the pre- and early post-exposure time points, we will assess the concentration of a range of known Damage-Associated Molecular Pattern molecules (DAMPs), including High mobility group box 1 (HMGB1), the S100 Calcium Binding Protein A1 (S100A1), and the alarmins, interleukin-33 (IL-33), and Thymic stromal lymphopoietin (TSLP). In addition, we will also quantify the concentration of DNA complexes associated with myeloperoxidase (MPO-DNA) as a marker of NETosis. At the pre- and 24 h post-exposure time points, a panel of Th1, TH2, and Th17 cytokines will be examined using Cytometric Bead Array (CBA) technology (BD Biosciences^®,^ New Jersey, NJ, USA) as follows: IL-10, IL-6, IL-1β, IL12p70, IL-8, IL-4, IL-2, Tumour Necrosis Factor alpha (TNFα), Interferon-gamma (IFN-γ), Granulocyte-macrophage colony-stimulating factor (GM-CSF).

The lavage samples at the pre- and immediately post-exposure time points will also be analysed for a range of metals and metalloids to act as source-specific exposure biomarkers as follows: Sb, Ba, and Cu, (brake wear). Zn (tyre wear), Fe, Mn, Mo (mechanical abrasion), Fe, Ca (road dust resuspension), and V, Ni and Cr as markers of diesel exhaust emissions. Analyses will be performed after nitric acid digestion using Inductively coupled plasma mass spectrometry (ICP-MS).

#### 2.9.3. Urine Samples

Urine samples will be collected by participants at home on the morning of site visits and during follow-up assessments the day after exposure. Urine samples will be aliquoted into clean 1ml polypropylene tubes and frozen at −80 °C for later analysis and biobanking.

### 2.10. Pre-Exposure Personal Air Pollution Monitoring:

To estimate any pre-testing exposure to PM, participants will be given a personal air quality monitor (Flow2, Plume Labs, Paris, France) to wear for three days prior to the exposure visits to highlight any significant and possibly confounding exposure to high PM. Indications of very high exposure to PM prior to testing may be used to exclude participants from analysis. In addition, we will examine longer-term area level exposures based on PM_2.5_, PM_10_, NO_2,_ and O_3_ concentrations measured at urban background monitoring stations in London. Background monitoring stations will be selected from sites within London’s Air Quality Network (https://www.londonair.org.uk/LondonAir/Default.aspx, accessed on 1 March 2023), based on their proximity to the residential address of the participants.

### 2.11. Exercise Intensity

To account for changes in intensity during exercise, participants’ heart rate will be monitored for the duration of the exercise exposure using polar heart rate monitors (Polar H9). Data at minute five and minute fifteen of each exercise bout will be averaged to assess for heart rate drift during the assessment. Borg scale ratings of perceived exertion will be collected and assessed for the same reason.

### 2.12. Common Allergy Assessment

An additional 5 mL blood sample will be collected to assess a full blood count, including eosinophil levels and specific immunoglobulin E levels across a panel of common aeroallergens to be included as covariates.

### 2.13. Symptoms Surveys

Participants will complete an Asthma Control Test (ACT) [26], an Asthma Quality of Life Questionnaire (AQLQ) [33], and general symptom questionnaires at the baseline, and will be asked to complete a daily asthma questionnaire three days before and three days after each exposure [34].

### 2.14. Air Pollution Measurements and Characterisation of Non-Exhaust Emissions

The air quality monitoring stations comprising of systems defined in Table 3 will provide measurement data according to the data quality assurance procedures described below; they will also enable a robust determination of the sources of exhaust, non-exhaust, and other urban and regional sources of PM_10_ and PM_2.5_.

Mass concentrations of PM_10_ and PM_2.5_, oxides of nitrogen (NO, NOx, NO_2_), and the ozone will all use reference measurement techniques supported by standardised operational and quality assurance approaches employed by the London Air Quality Network (www.londonair.org, accessed on 1 March 2023) or the UK government’s Automatic Urban and Rural Network (https://uk-air.defra.gov.uk/networks/network-info?view=aurn, accessed on 1 March 2023). Elemental composition will be measured at 1 hourly time resolution using ED-XRF [35]. To maximise the size and chemical composition information, PM_2.5_ and PM_10_ will be collected on alternate hours using a switching valve, and intermediate hours are interpolated, as described by Manousakas et al., 2022 [36]. The Aerosol Chemical Speciation Monitor [37] will measure the non-refractory composition of aerosols [38], and black carbon will be measured using an Aethalometer.

The road surface conditions will be collected from surface sensors at Marylebone Road (DRS511, Vaisala, Finland) and using a camera at White City Flyover (DSC111, Vaisala, Finland), which will be used to differentiate between wet and dry road surface conditions, as per [39].

### 2.15. Source Apportionment

The equipment described will identify sources of exhaust and non-exhaust concentrations via quantifying and source apportioning the high-time resolution chemical composition measurements described above in order to produce time series and factor profiles of source emissions in organic aerosols and elemental compositions. Using high-time resolution data in this way addresses many of the challenges faced when quantifying the different sources in the urban environment, as the higher time resolution yields a greater variability from changing emission characteristics and boundary layer dynamics and reflects short-lived events which are obscured by the long sampling time of filter-based approaches [40]. This variability is fundamental in improving the performance of multivariate statistical approaches, which are used to estimate the source contributions and fingerprints to solve a mass balance. Positive Matrix Factorisation (PMF) [41] is the most commonly used of these, and the solutions describe the complex, time-dependent aerosol composition as a linear combination of factor profiles and their contributions. It has been used extensively for the identification of heavy metals, water soluble and carbonaceous sources [42], and for high-time resolution elemental data from the Xact [43,44,45,46,47,48,49,50].

PMF will be applied to the PM_10_ and PM_2.5_ Xact data and ACSM data from all 3 stations using the Source Finder software 9.5.3 (Datalstica, Villigen, Switzerland) to provide hourly source factor time series and factor profiles for the duration of the exposures. For the ACSM data, the SoFi software has been used in many recent studies to provide high-quality source information using a priori information (both source profiles and time series) and bootstrap resampling approach by following a standardised source apportionment protocol from Crippa et al. [51] and Chen et al. [52], which is now widely used in the source apportionment community [53,54,55,56,57]. Recent work on data from Zurich [36] has shown that source apportionment of the XACT PM_10_ and PM_2.5_ data can be used to quantify the emissions from a range of sources in urban background locations.

Separating the tyre and road wear sources is not straightforward as they are internally mixed and generated by the same frictional forces. To help understand some of these processes in greater depth, additional size fractionated measurements of polymers are planned at Honor Oak Park and Marylebone Road during 2022/23 under a separate project (NERC, ‘Understanding UK airborne microplastic pollution: sources, pathways and fate’—NE/T007605/1).

High volume PM_2.5_ samples (DHA-80, Digitel Elektronik AG, Volketswil, Switzerland) will also be collected during the exposure periods and made available for additional laboratory analysis.

## 3. Data Analysis

### 3.1. Statistical Methods

The main health outcome variables that will be considered as dependent variables are as follows: FEV_1_ and FVC (differences between immediately, 24 h, and 3 days post-exposure and pre-exposure measurements); oscillometry and FeNO (same differences); asthma symptoms (yes/no) 24 h and 3 days post-exposure; and immune mediator concentrations HMGB1 and IL6 from nasal lavage (differences between 24 h and 3 days post-exposure from pre-exposure values). The exposure covariates will be the integrated average concentrations of the various pollutants and/or their sources during the exposure period. The results of the source apportionment will be used to evaluate the proportion of PM_2.5_ and PM_10_ mass related to each NEE and exhaust source. Additionally, the PM_10_ and PM_2.5_ total mass will be used alternatively to check the main associations as a priori expected. Potential confounders will be included, such as meteorological variables including temperature, relative humidity, and gaseous pollutants during exposure. Adjustments for heart rates will additionally be added. Individual characteristics will also be considered, such as age, sex, the severity of asthma, asthma control, quality of life, alcohol consumption, smoking, and allergies.

Mixed effects regression modelling (as appropriate for the distribution of each health outcome variable) will be used to estimate the effects of different exposure variables on the range of acute response endpoints measured in the current study. A key time point for each outcome measure has been selected, based on previous observations in controlled exposure studies or known temporal profiles of injury and inflammation (pre, 24 h post) following exposures to inhaled toxicants (pre, post). The analysis will begin with single exposure covariate models for every source type, which will be used to build a multi-exposure model containing all source types. The effects of thew different exposures will be compared based on statistical significance and the effect size for selected exposure contrasts, such as a pre-defined change in the concentration of the pollutant or per interquartile range. A model will be applied contrasting the three sites (as categorical variables) to explore whether the profile of each site as a whole is important for the examined health effects. We will use R (version 4.1.3 or newer; R Development Core team) for the application of mixed models using functions from the lme4 and nlme packages.

Missing data will not be imputed. Likely causes of missing data are technical issues associated with health assessments in field environments. Due to the impact of the technique on the performance with spirometry in particular, missing data are likely. During their pre-testing visit, researchers will ensure that participants can complete all testing and sampling before they visit a testing site. The study sample size has been increased by 20% to accommodate for missing data. Participants will also be given the opportunity to reschedule a visit if needed.

### 3.2. Ethics, Safety, and Dissemination

The study is sponsored and managed by QMUL and supported by Imperial College London. This study has received HRA approval following the favourable opinion from North West Scotland National Health Service Research Ethics Committee (NHS REC) (IRAS number: 320784). The study will be conducted according to the principles of the Helsinki Agreement (2013).

### 3.3. Governance

An Independent Steering Committee (ISC) chaired by Prof Flemming Cassee will monitor and advise the study’s conduct and progress on behalf of the sponsor and the HEI. The ISC will meet with the Chief Investigator and study team three times during the two-year study. At least one member of the AUKCAR PPI group will be included in the meetings.

The study is sponsored by QMUL (IRAS number: 320784) and is registered at clinicaltrials.gov. The monitoring and independent oversight of data are carried out by an independent assessor employed by QMUL and the audit team from the Health Effects Institute.

### 3.4. Consent

After discussing the study with the Principal Investigator, participants will have at least 24 h to consider participation. To confirm consent, participants have the option of emailing signed consent forms to the Principal Investigator or signing consent forms at the first study visit.

### 3.5. Data Management

Survey Lab will be used to store electronic case report forms (CRFs). All data will be held on GDPR-compliant databases with integrated data validation checks and audit trails. All collected data will be held on backed-up encrypted servers. Any data transfers between ICL and QMUL will be performed via Secure File Transfer Protocols. Paper records, CRFs, consent forms, and recruitment logs will be held locally in double locked locations at QMUL.

### 3.6. Confidentiality

We will follow the best practice guidelines provided in Standard Operating Procedures (SOPs) by the Pragmatic Clinical Trials Unit (PCTU) at QMUL. Paper records will be stored securely in locked filling cabinets in password-locked rooms in the pass-protected Centre for Primary Care. Electronic records will be stored in a password-protected study database on a secure server in the Centre for Primary Care. In the study, the database and personal details (name, address, date of birth) will be kept separate from the research data, which will be identified by a unique study reference number. In the tables of data, participants will only be identified by number, not by initials or name. Data management procedures will be completed in compliance with the GDPR and trial regulations. Survey-reported data will be stored in the QMUL data safe haven, where data will be held in a UK server and access will be facilitated via two-factor authentication. Survey software with integrated data validation checks and audit trails will be used to record study data. Any data transfers between QMUL and Imperial College London will be completed via an encrypted Secure File Transfer Protocol. All data will be backed up weekly to ensure that the data are safeguarded from accidental loss. The data will only be accessed and used by those members of the research team at QMUL and Imperial College London and the representatives of the sponsor who have been granted permission.

### 3.7. Record Retention and Archiving

In accordance with the UK Policy Framework for Health and Social Care Research, research records will be kept for 20 years after the study has been completed, while personal records will be stored for one year after study completion. At the completion of the study, data will be moved to a trusted archive centre. At the end of the retention period, data will be destroyed in accordance with the best practice guidelines at the time of destruction.

### 3.8. Adverse Events Reporting

A risk and adverse events register will be maintained for the duration of the study. Accidents will be discussed with the Chief Investigator and reported promptly to the sponsor as per Good Clinical Practice regulations and reviewed at study Project Management Group meetings.

### 3.9. Dissemination Plan and Project Outputs

Results arising from this study will be reported to the relevant funding agencies, scientific community, and stakeholder groups through the following means:(1)Webinars on websites of our institutions to provide more detailed summaries of results, with downloads of key documents.(2)Presentations, especially to London organisations including the GLA, councils, and health and wellbeing boards.(3)Peer-reviewed publications.(4)Presentations at national and international conferences.

## 4. Discussion

Internationally, policies to reduce pollution from traffic have predominately focused on exhaust emissions. This has been motivated by evidence of health effects associated with the proximity of populations to traffic sources and with proxies of primary exhaust emissions: black/elemental carbon, NO_2,_ and PNC [58]. However, as policy actions have driven reductions in exhaust emissions, attention has turned to NEEs from vehicles arising from the wear of brakes, tyres, and the road surface. These are currently unregulated and often exceed the total mass contribution of exhaust emissions in roadside and background PM_2.5_ and PM_10_. The significance of these sources will increase with the greater penetration of non-internal combustion engines into the vehicle fleet as countries transition away from fossil fuels to meet their NetZero commitments. There is therefore an urgent need to establish the contribution of these sources to the adverse health effects attributed to ambient PM. Such information will be essential for the development of evidence-based policies to protect public health over the coming decades and to inform innovation within the automotive sector to reduce the risk of non-exhaust traffic emissions. While the knowledge of the sources and compositional signatures of non-exhaust particles is becoming more established, significant uncertainty remains and there is limited information on their relative toxicity and health impacts.

Asthma is globally the most prevalent long-term condition, strongly impacted by air pollution, affecting adults and children, with major morbidity disproportionately affecting disadvantaged and minoritised ethnic populations. The asthmatic airway provides an exquisitely sensitive model to assess the health impacts of non-exhaust emissions. The use of a quasi-experimental human real-world crossover design employed in this study allows for the study of the health effects of non-exhaust emissions and to explore the hypothesis that clinically important adverse acute asthmatic responses are driven by non-exhaust components within coarse mode particulate matter.

Collectively, this study will provide us with valuable information on the health effects of NEE components within ambient PM_2.5_ and PM_10_, whilst establishing a biological mechanism to help contextualise the current epidemiological observations. Moreover, this work will establish key air quality and health variables to support larger epidemiological work exploring the health impacts of non-exhaust air pollution that will ultimately contribute to the national air quality guidelines.

## Figures and Tables

**Figure 1 ijerph-21-00895-f001:**
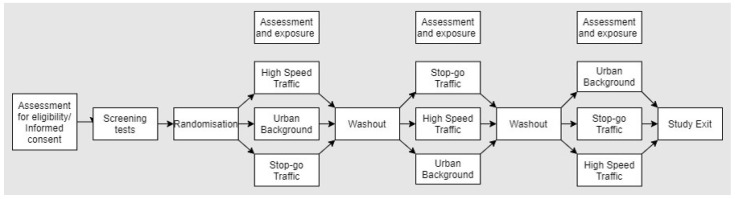
Diagram of participant flow through IONA study.

**Figure 2 ijerph-21-00895-f002:**
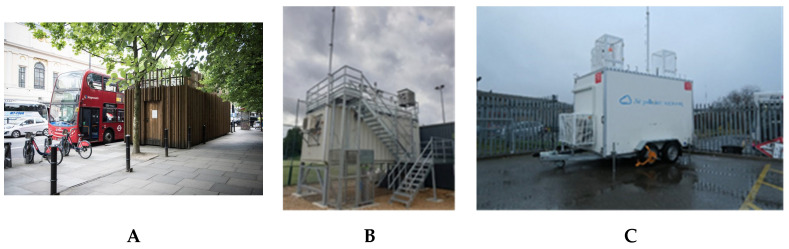
Air quality supersites: Marylebone Road (**A**), Honor Oak Park (**B**), mobile measurement station (not in location) (**C**).

**Table 1 ijerph-21-00895-t001:** IONA study inclusion and exclusion criteria.

Inclusion Criteria	Exclusion Criteria
1. Doctor-diagnosed asthma starting on or before age 16 years	1. Current or previous tobacco smoking, or living with a smoker
2. Prescribed regular inhaled corticosteroid medication	2. BMI > 30
3. Able to use a static exercise bicycle for the study duration	3. Asthma hospitalisation within 12 months
	4. Three or more asthma episodes requiring oral corticosteroid medication within 12 months
	5. Other major lung disease
	6. Chest surgery within 6 months
	7. Unable to give informed consent
	8. Occupational exposure to PM or high levels of air pollution ^a^
	9. Under the age of 18
	10. Individuals at any stage of pregnancy
	11. Currently breast feeding

^a^ For the purposes of this study, occupational exposure is defined as people who work as taxi drivers, couriers, waste removal drivers, and utility services drivers. If other occupations with potentially high exposure to air pollution approach the study, the study team will discuss the eligibility of the occupation within the project management group, comprising experts in air pollution and asthma management.

**Table 2 ijerph-21-00895-t002:** SPIRIT schedule for IONA study.

	IONA: Schedule of Assessment and Exposure							
	Pre-Study	Initial Assessment	3-Day Pre	Pre	Exposure	Post	24 h Post	3-Day Post
		Clinic Room	Self-Assess	Field Lab	Field Lab	Field Lab	Clinic Room	Self-Assess
Consent	x							
Exercise test		x						
FeNO		x		x	x	x	x	
Oscillometry		x		x	x	x	x	
Spirometry		x		x	x	x	x	
Asthma control test		x						
Asthma quality of life test		x						
Asthma symptoms			x				x	x
Venepuncture				x			x	
Nasal lavage				x		x	x	
Urine Sample				x			x	
Heart rate					x			

**Table 3 ijerph-21-00895-t003:** Air quantity station measurement configuration.

Measurement	High-Speed Continuous Traffic	Stop–Go Traffic	Urban Background
	White City	Marylebone Road	Honor Oak Park
PM_10_ and PM_2.5_ Mass Concentration	Optical Particle Counter, Fidas 200E, Palas, Karlsruhe, Germany		
NO_2_	T200 Chemiluminescence NO/NOx, or N500 Cavity Attenuated Phase Shift, Teledyne, Thousand Oaks, CA, USA		
O_3_	T400, UV Absorption, Teledyne, USA		
40 elements using XRF (Al, Si, P, S, Cl, K, Ca, Ti, V, Cr, Mn, Fe, Co, Ni, Cu, Zn, Ga, Ge, As, Se, Br, Sr, Y, Zr, Mo, Pd, Ag, Cd, In, Sn, Sb, Te, Ba, La, Ce, Pt, Hg, Tl, Pb, Bi) in PM_2.5_ and PM_10_		Xact 625i, Cooper Environmental Services, Edgartown, MA, USA	
23 elements using XRF (Si, S, Cl, K, Ca, Ti, V, Cr, Mn, Fe, Ni, Cu, Zn, As, Se, Sr, Mo, Cd, Sb, Ba, Ce, Pt, Pb) in PM_2.5_ and PM_10_	Xact 625, Cooper Environmental Services, USA		
Organic Mass, NO_3_, SO_4_, NH_4_ in PM_2.5_ or PM_1_	Aerosol Chemical Speciation Monitor (ACSM), Aerodyne Research Inc., Billerica, MA, USA		
Black Carbon in PM_2.5_	Aethalometer AE33, Magee Scientific, Berkeley, CA, USA		

## Data Availability

A copy of the study data will be held on the HDRUK BREATHE secure data hub. BREATHE’s mission is ‘Better respiratory health through better connected data’.

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
