# Peer review of "Assessing the Impact of Non-Exhaust Emissions on the Asthmatic Airway (IONA) Protocol for a Randomised Three-Exposure Crossover Study"

_ijerph, 2024, doi:10.3390/ijerph21070895_

Round 1
Reviewer 1 Report
Comments and Suggestions for Authors
This paper presents a protocol for studying and assessing the effects of Non-exhaust Emissions (NEE) on the lung function and airway immune status of asthmatic adults. This is a topical issue as the impact of NEE on human health is still under investigated.
The manuscript is rather well written and presented.
Main comments:
1 - The paper contains few typos, please check and cleanup
2 - What is the duration of the study? I might had miss it, so state it clearly
3 - Sample size: the authors mention that they are willing to recrut 48 adults of age > 18 years. The argument presented in this subsection is not clear enough on how the authors derived this number. In addition, I am concern by how small is that number in regard with the number of variables involved in the study. One would expect a lager number of sample size at least in the case of constructing a biobank.
4 - All collected data should be synthesized in Tables. Please, provide some.
5 - In addition, it would be very useful for such a study to construct a "IONA" database. In that case, I encourage the authors to work out a template table showing both the content of the database and how data are stored.
Author Response
Dear Reviewer,
Thanks for taking the time to review the manuscript. We believe your comments have improved the manuscript. We have responded to your comments below:
1 - The paper contains few typos, please check and cleanup
We have re-read and corrected any typos we found.
2 - What is the duration of the study? I might had miss it, so state it clearly
Between May and October 2023.
Mentioned only once para 2 in page 4. We have therefore reiterated the timeline/duration in:
Recruitment section page 6
Exposures section page 9.
3 - Sample size: the authors mention that they are willing to recrut 48 adults of age > 18 years. The argument presented in this subsection is not clear enough on how the authors derived this number. In addition, I am concern by how small is that number in regard with the number of variables involved in the study. One would expect a lager number of sample size at least in the case of constructing a biobank.
Our a priori power calculations showed that we required 30 participants at the point of analysis to achieve a 91.5% power. 90% power is broadly considered a suitable strength for analysis. This was based off mean changes and standard deviations in FEV1 measured by spirometry (IONA primary outcome variable is FEV1) from the Oxford St. exposure study conducted by McCreanor in 2007. A study conducted in an environment analogous to the proposed IONA study. IONA sample size is also of similar size to other PM exposure studies, such as RAPTES.
To ensure we had a sample size of 30 participants at point of analysis we chose to inflate the sample to 48 this accounted for a high dropout in participants the unknown air quality composition of the sites prior to testing and facilitated randomisation to a Williams design. Our analysis will also allow the inclusion of partial data from participants who did not attend all sites but are willing to allow their data to be included
While the primary outcome variable is at 91.5% statistical power we accept secondary outcome variables may not be sufficiently powered. This is typical of well controlled but small panel studies such as this, we shall ensure that our the findings and discussions in future publications which present data reflect this.
Regarding the biobank: the current state of science is not able to measure non tailpipe emissions in a method to support large epidemiological work. Small well controlled exposure panel studies such as IONA are vital to progressing pollutant source apportionment technique and health biomarker measurements to a stage where they can support large epidemiological work. We accept small panel studies such as this come with inherent risks and we shall discuss these appropriately when presenting analysis. Analysis for the samples in the biobank have already been proposed and grant applications are submitted. We are mitigating the small size of the biobank by linking IONA samples to other small well controlled panel studies to maximize contrast in variables of interest.
4 - All collected data should be synthesized in Tables. Please, provide some.
This is a protocol paper and does not present any data. Are you able to clarify what you would like us to present?
5 - In addition, it would be very useful for such a study to construct a "IONA" database. In that case, I encourage the authors to work out a template table showing both the content of the database and how data are stored.
We agree that a database is vital to support this project. We have two databases one containing PID and the other containing study data. An overview of the databases described in Page 12.
Our database is saved in our secure data safe haven. I have requested a template example from our data management team. However I have not been able to get it in time to meet the journal's deadline for a response to reviewers. I shall include it in the next round if required.
Reviewer 2 Report
Comments and Suggestions for Authors
Journal:International Journal of Environmental Research and Public Health
Title: Assessing the Impact of Non-exhaust Emissions on the Asth-matic Airway (IONA) Protocol for a randomised three exposure crossover study (protocol)
Comments: Reduction in exhaust emissions have been driven worldwide to prompt the control of health effects posed by traffic pollution. However, non-exhaust emissions remain unregulated and underexplored about the associated health impacts. The significance of these sources will increase with greater penetration of non-internal combustion engines into the vehicle fleet as countries transition away from fossil fuels to meet their NetZero commitments. It’s imperative to assess the relative toxicity and health impacts of non-exhaust emissions, especially in vulnerable groups such as asthma adults. This protocol described the study design, the schedule of assessment and exposure, recruitment criteria, exposure assessment, respiratory function assessment, sample collection, and air pollution measurements. The design is explicit, and this study is meaningful for the public health. It’s recommender to be accepted after minor revision.
(1) According to the abstract in current version, it’s hard for readers to get the useful information about this study. It’s recommended that the authors added the descriptions about the panel design, exposure experiments procedure, biomarkers measurements, samples collections, and questionnaire survey.
(2) The inclusion and exclusion criteria narrated in the text was confused a little bit. It’s better to establish a table to summarize the criteria.
Author Response
Dear reviewer
thank you for your positive comments about the study and manuscript we have included responses to your direct comments below
- According to the abstract in current version, it’s hard for readers to get the useful information about this study. It’s recommended that the authors added the descriptions about the panel design, exposure experiments procedure, biomarkers measurements, samples collections, and questionnaire survey.
We agree I have made changes to the abstract including key bits of what you suggest. To ensure we meet word count I have greatly abridged the background section. We agreed this fix the section much more useful to reader
panel design: “A randomised three condition crossover panel design”
exposure experiments procedure: “using an 2.5 hour intermittent cycling protocol”
biomarkers measurements, samples collections, and questionnaire survey:
Lung function will be monitored by oscillometry, fractional exhaled nitric oxide and spirometry (primary outcome). Biomarkers of inflammation and airborne metal exposure will be measured in the upper airway via nasal lavage. Symptom responses will be monitored by questionnaires.
Sources of exhaust and non-exhaust concentrations will be established using source apportionment via positive matrix factorisation of high-time resolution chemical measures conducted at the exposure sites.
- The inclusion and exclusion criteria narrated in the text was confused a little bit. It’s better to establish a table to summarize the criteria.
We have changed to a table as suggested
Reviewer 3 Report
Comments and Suggestions for Authors
Dear authors
I read your manuscript with great interest. Below you will find some doubts and suggestions that I hope can contribute to the final version.
(the version of the manuscript did not have the lines, so I indicate the pages)
Page 2. When you say: “…road dust, represent a greater proportion of roadside PM by mass than direct exhaust emissions (Piscitello et al, 2021)”, could this depend on the quality of the pavements, the proportion of newer and more modern vehicles?
Page 2 (at the end): I think you have an error “The contrasting traffic profiles increase the contrast in (?) PM10 and PM2.5 from brake”
Page 2 (at the end): do you include former smokers? It's not clear and is not the same.
Page 3: please review these sentences: “If other occupations with potentially high exposure to air pollution approach the study. The study team will discuss eligibility of the occupation within the project management group, compris-ing of experts in air pollution and asthma management.”
Page 4: You say that “…while remaining below the first ventilation threshold to reduce variation in ventilatory drift”. I think that this means that participants will have to be in good physical condition, as I mention again later.
Page 7: I think you mean pre and post-bronchodilator spirometry?
Page 7: “…estimated Vo2max which will in turn will be used to calculate a target watt equivalent for 60% estimated Vo2max. “. As I mentioned before, this is suitable for participants in good physical condition. Otherwise, at 60% the AT may have already been exceeded.
Page 7 (spirometry): Operators are going to have any prior training? To avoid variability due to the spirometry technique (which is not guaranteed just by using the guidelines). For example, for an accurate FEV1 a full inspiratory capacity is mandatory.
Page 11: correct “oscillometery”
Kind regards
Author Response
Dear reviewer,
Thank you for your kind words and your thoughtful comments we have provided responses to all of your comments below, we believe they have improved the manuscript.
Page 2. When you say: “…road dust, represent a greater proportion of roadside PM by mass than direct exhaust emissions (Piscitello et al, 2021)”, could this depend on the quality of the pavements, the proportion of newer and more modern vehicles?
Thanks for this comment. Yes many factors impact road dust. Including the environment, weather, cleaning protocols and systems, vehicle types (newer cars tend to be heavier and therefore produce greater non exhaust emissions), tyre compounds and many other things. This statement for dust representing a greater proportion of roadside mass is a general one and remains true regardless of environment and vehicle fleet composition.
Page 2 (at the end): I think you have an error “The contrasting traffic profiles increase the contrast in (?) PM10 and PM2.5 from brake”
(?) Removed
Page 2 (at the end): do you include former smokers? It's not clear and is not the same.
We excluded former smokers. We have amended the inclusion criteria to clarify this
“Current or previous tobacco smoking, or living with smoker”
Page 3: please review these sentences: “If other occupations with potentially high exposure to air pollution approach the study. The study team will discuss eligibility of the occupation within the project management group, compris-ing of experts in air pollution and asthma management.”
Corrected as suggested.
Page 4: You say that “…while remaining below the first ventilation threshold to reduce variation in ventilatory drift”. I think that this means that participants will have to be in good physical condition, as I mention again later.
We agree. It is plausible that some less trained participants may crossed the threshold as such we have altered the wording two soften the statement to “minimizing the likelihood of ventilatory drift”
Page 7: I think you mean pre and post-bronchodilator spirometry?
Yes. Corrected as suggested.
Page 7: “…estimated Vo2max which will in turn will be used to calculate a target watt equivalent for 60% estimated Vo2max. “. As I mentioned before, this is suitable for participants in good physical condition. Otherwise, at 60% the AT may have already been exceeded.
Agreed and have softened wording as per previous.
Page 7 (spirometry): Operators are going to have any prior training? To avoid variability due to the spirometry technique (which is not guaranteed just by using the guidelines). For example, for an accurate FEV1 a full inspiratory capacity is mandatory.
Agreed. All the trainings performed either by a senior respiratory physiologist or by trained technicians under their supervision I have included a line to reflect this.
“All testing was performed by trained operators under the supervision of a senior respiratory physiologist.”
Page 11: correct “oscillometery”
Corrected.
Round 2
Reviewer 1 Report
Comments and Suggestions for Authors
No further questions
Author Response
Thanks for your comments and support.
Reviewer 3 Report
Comments and Suggestions for Authors
Dear authors
I acknowledge the changes that have been made, but I still have doubts regarding the important issue of VO2 max %. In a normal subject with good physical condition, the AT occurs at approximately 60% of the predicted VO2max. As we know, in a normal response to exercise, VO2 increases linearly with HR. However, in an subject without disease but with poor physical condition, HR increases more rapidly and AT is reached earlier (the response is similar to that of a subject with cardiac limitation due to mild heart disease). It does not seem to me that this issue is reflected in your protocol because you wrote that “Participants will exercise on cycle ergometers … at corresponding power out-put of 60% of their estimated Vo2max “, unless, as I mentioned earlier, all participants will have good physical condition
kind regards
Author Response
60%Vo2max-
Thanks for carefully explaining your comment. I was a bit slow on the uptake!
I agree with yours comments.
I believe that our inclusion criteria (Namely, the requirement to cycle intermittently for 2.5 hours) requires participants to be of at least good fitness.
I have included a check in the protocol to confirm that there is a linear relationship between vo2 and HR:
page 8: Vo2 and heart rate scores will be inspected to ensure a linear relationship during the exercise protocol to ensure accuracy of estimate of exercise intensity.